# Single-Cell Sequencing Reveals an Intrinsic Heterogeneity of the Preovulatory Follicular Microenvironment

**DOI:** 10.3390/biom12020231

**Published:** 2022-01-29

**Authors:** Huihua Wu, Rui Zhu, Bo Zheng, Guizhi Liao, Fuxin Wang, Jie Ding, Hong Li, Mingqing Li

**Affiliations:** 1Center of Reproduction and Genetics, The Affiliated Suzhou Hospital of Nanjing Medical University, Suzhou Municipal Hospital, Suzhou 215002, China; whh_200909@126.com (H.W.); summerrui2006@aliyun.com (R.Z.); bozheng@njmu.edu.cn (B.Z.); liaoguizhi@163.com (G.L.); cathywangfuxin@163.com (F.W.); dj95028@163.com (J.D.); 2Laboratory for Reproductive Immunology, Hospital of Obstetrics and Gynecology, Shanghai Medical School, Fudan University, Shanghai 200080, China

**Keywords:** scRNA-seq, ovulation, preovulatory follicle, granulosa cell, macrophage

## Abstract

The follicular microenvironment, including intra-follicular granulosa cells (GCs), is responsible for oocyte maturation and subsequent ovulation. However, the functions of GCs and cellular components of the follicular microenvironment in preovulatory follicles have not been extensively explored. Here, we surveyed the single-cell transcriptome of the follicular microenvironment around MII oocytes in six human preovulatory follicles in in vitro fertilization. There were six different cell types in the preovulatory follicles, including GCs and various immune cells. In GCs, we identified nine different functional clusters with different functional transcriptomic profiles, including specific clusters involved in inflammatory responses and adhesive function. Follicular macrophages are involved in immune responses, extracellular matrix remoulding and assist GCs in promoting the oocyte meiotic resumption. Interestingly, we observed that the specific terminal state subcluster of GCs with high levels of adhesive-related molecules should result in macrophage recruitment and residence, further contributing to an obvious heterogeneity of the immune cell proportion in preovulatory follicles from different patients. Our results provide a comprehensive understanding of the transcriptomic landscape of the preovulatory follicular microenvironment at the single-cell level. It provides valuable insights into understanding the regulation of the oocyte maturation and ovulation process, offering potential clues for the diagnosis and treatment of oocyte-maturation-related and ovulation-related diseases.

## 1. Introduction

The ovary, an important reproductive organ, serves as the source of oocytes and is the major supplier of the steroid sex hormone in women [1]. One major functional component of the ovary is the ovarian follicle. Follicles are composed of somatic cells, such as granulosa cells (GCs) and other cell types that exhibit both endocrine and developmental functions. The cells in the follicular fluid and the cumulus granulosa cells (CCs) around the oocyte constitute the follicular microenvironment [2] (Figure 1A). In preovulatory follicular cells, cellular signalling cascades can be activated by a luteinizing hormone (LH) surge or human chorionic gonadotropin (HCG) administration, activating the expression of a number of genes to promote oocyte meiotic resumption, remodel the follicle structure, and prepare to support the luteal function [3,4,5]. Therefore, it is of great importance to deeply understand the preovulatory follicular environment after LH/HCG and its potential roles in ovulation. 

In the follicular microenvironment, the intra-follicular GCs are responsible for the dynamics of ovarian functions including the balance of signals necessary for oocyte maturation and subsequent ovulation. The interactions between GCs and oocytes via gap junctions regulate the oocyte growth and final maturation [6]. After the LH surge, the regulation of the oocyte meiotic maturation requires the activation of the epidermal growth factor (EGF) network in GCs [7]. In mouse GCs, the EGF-like peptides amphiregulin (AREG) and epiregulin (EREG) decrease the *NPPC* transcription and the cGMP level [8,9], the two major elements inhibiting oocyte meiotic resumption [10,11,12]. Meanwhile, the steroidogenesis in GCs transforms from the estradiol (E2) synthesis into the progesterone (P) synthesis. Progesterone interacts with progesterone receptors (PGRs) and then regulates the expression of many genes involved in tissue remodelling and inflammatory responses [13,14]. Being a critical cell type in folliculogenesis, GCs divide into two major cell subtypes: mural granulosa cells (MGCs) and CCs in the selectable follicle state, whose functional differences have been described in previous studies [15,16]. However, the diverse functions and heterogeneity of GCs located in different compartments of preovulatory follicles remain largely unknown.

Furthermore, oocyte development and maturation are regulated by the microenvironment of follicles. Follicular fluid-derived cells isolated from follicles aspirated at the oocyte retrieval from in vitro fertilization (IVF) patients have been described as a heterogeneous cell population [17], including GCs and epithelial cells, as well as various immune cells [18,19]. The cytokines and chemokines derived from immune cells play important roles in ovarian functions, participating in follicle growth, oocyte maturation, ovulation, and luteinization [20,21]. However, the heterogeneity of immune cells and cell–cell communications between immune cells and GCs in preovulatory follicles in IVF are still unclear. 

To investigate the heterogeneous cell population of the follicular microenvironment in the IVF cycle and their specific functions in preovulatory follicles, we adopted the cutting-edge single-cell RNA sequencing (scRNA-seq) technique [22], and delineated the first single-cell transcriptomic landscape of the microenvironment around MII oocytes in preovulatory follicles in IVF. Our results identified different functional clusters of granulosa cells as well as other cell types in preovulatory follicles, revealed the dynamics of specific genes in GCs during ovulation, and disclosed the potential cellular interactions between major cell types, including various immune cell types. It provided a detailed and comprehensive investigation of the microenvironment in preovulatory follicles and brought novel insights into understanding the ovulation process.

## 2. Material and Methods

### 2.1. Participants

For this study, six follicular fluid and CC samples were collected from six patients undergoing oocyte retrieval and intracytoplasmic sperm injection (ICSI) at the Reproduction and Genetics Centre of the Affiliated Suzhou Hospital of Nanjing Medical University. These patients, with an average age of 29.67 ± 4.08, received the ICSI treatment for male-factor infertility, and had a regular menstrual cycle. The exclusion criteria were as follows: (1) patients with an ovulatory disorder or diminished ovarian reserve; (2) patients with tubal factors; (3) patients with endometriosis and adenomyosis; and (4) patients with BMI ≥ 25 kg/m^2^. This research was approved by the Medical Ethical Committee of the Affiliated Suzhou Hospital of Nanjing Medical University (code number 2020009). Signed informed consent was obtained from all patients who participated in this research.

For the patients, recombinant follicle stimulating hormone (Gonal-F, Merck Serono, Darmstadt, Germany or Puregon, NV Organon, Oss, The Netherlands) treatments were started on day 2 of the menstrual cycle, or 30–35 days after the gonadotropin-releasing hormone (GnRH) agonist (Leuprorelin Acetate Microspheres for injection, Lizhu, China) injection, with the option to adjust the dose according to the response after a 4-day stimulation. In the GnRH antagonist protocol, when the leading follicle reached 13–14 mm in diameter, the GnRH antagonist (Orgalutran, NV Organon, Oss, The Netherlands) 0.25 mg daily was administrated until the day of HCG administration. Then, HCG (Ovidrel, Merck Serono, Darmstadt, Germany) only, or HCG (chorionic gonadotropin for injection, Lizhu, China) with triptorelin acetate injection (Decapeptyl, Ferring, Kiel, Germany), were administrated when one or two leading follicles reached 18 mm in diameter. Transvaginal ultrasound-guided oocyte retrieval was performed 34–36 h later. 

### 2.2. Biological Sample Collection

The first punctured follicle with a diameter of 18–20 mm and containing MII-stage cumulus–oocyte complexes (COCs) from each patient was used. The follicular fluid was clear and free of visible blood contamination and the follicular fluid was collected individually. The cumulus cells from the COCs were dispersed by gentle pipetting in 1% hyaluronidase enzyme and washed with PBS. Then, they were resuspended in GIVF (Vitrolife, Sweden). The follicular fluid cells and cumulus cells isolated from the same follicle were immediately used for scRNA-seq. The MII-stage oocyte was fertilized by ICSI and cultured to the blastocyst stage separately.

### 2.3. Sample Preparation and Single-Cell RNA Sequencing 

The collected cells were centrifuged, and then the cell pellets were resuspended in PBS (HyClone, Marlborough, MA, USA). GEXSCOPE Red Blood Cell Lysis Buffer (Singleron Biotechnologies, Nanjing, China) was added to the cell suspension to remove the red blood cells. Then, the mixture was centrifuged and resuspended in PBS. Trypan blue exclusion test was used to evaluate the cell viability [23]. We counted the cells with a TC20 automated cell counter (Bio-Rad, Hercules, CA, USA), and prepared the cell suspension with a final concentration of 1 × 10^5^ cells/mL. 

Single-cell suspensions were loaded onto microfluidic devices. ScRNA-seq libraries were constructed on the basis of the Singleron GEXSCOPE^®^ protocol by the GEXSCOPE^®^ Single-Cell RNA Library Kit (Singleron Biotechnologies, Nanjing, China) and Singleron Matrix^®^ Automated Single-Cell Processing System (Singleron Biotechnologies, Köln, Germany). After being diluted to 4nM, individual libraries were pooled for sequencing. Libraries were then sequenced on an Illumina HiSeq X with 150 bp paired-end reads (Illumina, San Diego, CA, USA).

### 2.4. Primary Analysis of the Sequencing Data 

The primary analysis pipeline CeleScope was used to process the raw data (https://github.com/singleron-RD/CeleScope) (accessed on 22 December 2020). Briefly, read one of the sequencing data contains the cell barcode (CB) and the unique molecular identifier (UMI) and read two contains the gene information. Both were used for counting the expression levels of each gene in each cell. Low quality reads, poly-A tails and adaptor sequences were removed from raw reads by fastQC [24] and fastp [25]. After quality control, we used STAR (version 020201) to map the reads to the reference genome GRCh38 (Ensembl version 92 annotation) [26]. According to the results of the gene quantification performed by featureCounts (version 1.6.2) [27], sequencing reads from the same gene, same CB, and same UMI were lumped together as the PCR duplicates. Expression matrix files for subsequent analyses were generated by only counting the number of UMIs for each gene in each cell.

### 2.5. Quality Control, Dimension-Reduction and Clustering

We retained cells with 200–5000 genes, UMIs less than 30,000, and a mitochondrial gene expression percentage of less than 50% for the following analyses. Due to the increased glucose consumption in GCs throughout oocyte maturation and the increased production of enzymes involved in cholesterol conversion in mitochondria [28,29], we chose a threshold of 50% for mitochondrial gene filter. The sample-wise violin plots for genes, UMIs and mitochondrial gene expression percentage were showed in Appendix A. After filtering, 14,592 cells, with a median number of genes of 2242 and a median UMI of 6312 per cell, were retained for downstream analyses. Functions from Seurat [30] v3.1.2 were used for dimensionality reduction and clustering. We used NormalizeData and ScaleData to normalize and scale the gene expression matrix. For the PCA analysis, the top 2000 variable genes were chosen by FindVariableFeatures [31]. Cells were separated into 23 clusters by FindClusters, by using the top 20 principle components and a resolution parameter of 1.2. For the clustering of GCs and macrophages, we set the resolution to 1.2 and applied the uniform mainfold approximation and projection (UMAP) algorithm to visualize cells in a two-dimensional space [32,33]. Batch effects between samples were removed for clustering by Harmony [34] (1.0) with the parameter group.by.vars set as ‘sample’.

### 2.6. Differentially Expressed Gene (DEG) Analysis

Based on the Wilcox likelihood-ratio test with default parameters [35], we utilized Seurat FindMarkers to identify DEGs. DEGs were selected only if genes were expressed in more than 10% of the cells of a cluster and had an average log (fold change) value greater than 0.25.

### 2.7. Cell Type Annotation

With the expression of canonical markers found in the DEGs and knowledge from the literature, we annotated the cell type of each cluster. Cells which positively expressed double canonical marker genes of major annotated cell types and clusters were identified as doublets and were removed. Heatmaps, dot plots and violin plots were generated by Seurat DoHeatmap, DotPlot and Vlnplot, respectively, displaying the expression of markers used to identify each cell type. 

### 2.8. RNA Velocity Analysis

Furthermore, we used an RNA velocity analysis to further analyse the differentiation of GCs. Firstly, the BAM file for each sample was transformed into a loom format file by velocyto (version 0.17.17) [36]. Secondly, the loom format file was used as an input for scVelo (0.2.3) with default parameters [37]. Thirdly, we projected the result to the UMAP plot generated by the consistency of Seurat visualization. 

### 2.9. Pathway Enrichment Analysis

In order to assist the comprehension of the potential functions of each cell type, the gene ontology (GO) enrichment analysis [38] and Kyoto Encyclopedia of Genes and Genomes (KEGG) analysis [39] were performed with the “clusterProfiler” R package [40] (3.10.1). Pathways with a adjust *p*-value (p_adj) less than 0.05 were considered significantly enriched.

As for the GSVA [41] pathway enrichment analysis, the average gene expression for each cell type/cluster was used as the input data. Gene ontology gene sets and immunologic signature gene sets were selected as the reference. The top significant pathways were visualized in heatmaps.

### 2.10. Cell–Cell Interaction Analysis 

According to the known ligand-receptor repository CellPhone DB (version 1.1.0) [42], cell-cell interactions between different cell types and clusters were predicted. We set the threshold of cells expressing a gene within each cluster as 0.1, the iteration number as 1000, and other parameters as default. Predicted interaction pairs with a *p* value less than 0.05 were regarded as significant, and subsequently used for downstream analyses.

### 2.11. Flow Cytometry

The patients participating in the flow cytometry experiment were selected with the same inclusive and exclusive criteria as for the scRNA-seq. After the follicular fluid and cumulus cells were washed with PBS, the red blood cell lysis buffer solution (Beckman Coulter) was added to the sediment for 10 min. Then, the solution was centrifuged at 500 g for 7 min and washed by PBS as soon as possible. Dissociated cells were washed and stained according to the manufacturer’s protocol with the following antibodies alone or in varying combinations: CD45 (368505, Biolegend), CD68 (333807, Biolegend), CD3 (300306, Biolegend), CD4 (344638, Biolegend), CD1C (331505, Biolegend), HLA-DR (307609, Biolegend), CD 15 (301907, Biolegend) and CD11B (101205, Biolegend). Stained samples were analysed by flow cytometry (Navios, Beckman Coulter Ireland, Ireland). The data analysis was performed using FCS Express 6 (Kaluza analysis, Beckman Coulter, Indianapolis, IN, USA). 

### 2.12. Immunofluorescence

The cells for immunofluorescence were isolated from the follicular fluid and COCs in a single preovulatory follicle using density gradient centrifugation at 2000 rpm for 30 min with a human lymphocyte separation medium (Ficoll, Hao Yao Biological Manufacture, Tianjin, China). Following centrifugation, the middle layer was collected and washed with PBS. A total of 10^4^ cells/well were seeded on 12 well plates in F12 (Gibco, Thermo Fisher, Beijing, China). After incubation for 24 h at 37 °C in a humidified atmosphere (95% air and 5% CO_2_), the non-adherent cells were removed from the wells by washing with PBS. Then, the adherent cells were assessed with immunofluorescence for the GC marker (STAR), macrophage marker (CD68), as well as macrophage subcluster markers (EREG, CTSD, DAB2 and S1000A8).

Immunofluorescence experiments were conducted as previously described [43]. Briefly, the cells were fixed with 4% paraformaldehyde (PFA) for 25 min, permeabilized with Triton X-100 (0.4% in PBS) for 25 min, incubated with the blocking buffer (10% bovine serum in PBS) for 1 h at room temperature (RT), and stained with primary antibodies (rabbit anti-StAR, Cst; mouse anti-CD68, Abcam; rabbit anti-DAB2, Abcam; rabbit anti-S100A8, SAB; rabbit anti-CTSL, Abcam; rabbit anti-CD68, Abcam, mouse anti-EREG, Santa Cruz) overnight at 4 °C. Then, the cells were incubated with secondary antibodies (Cy3-labeled goat anti-mouse IgG, Alexa Fluor 488-labeled goat anti-rabbit IgG) for 1 h at RT. Hoechst 33342 (Thermo Fisher Scientific) was used to stain the nuclear DNA. Fluorescent images were obtained on an inverted confocal microscope (Nikon A1, Nikon, Tokyo, Japan).

## 3. Results

### 3.1. Single-Cell Clustering and Cell Type Identification in Human Preovulation Follicles

To determine the full repertoire of cell types and gene expression programs present in human preovulation follicles, six samples containing follicular fluid and cumulus cells were collected for scRNA-seq from patients undergoing oocyte retrieval and ICSI. These patients received ICSI treatment due to male-factor infertility and had regular menstrual cycles. These six oocytes were all in the MII stage and individually fertilized by ICSI and cultured to the blastocyst stage (Appendix A**)**. 

A total of 15,865 cells from six samples were applied for the scRNA-seq. The median number of genes detected per cell were 2309 and the median UMI detected per cell was 6611 (Appendix A). After quality control, 14,592 cells from these six samples were retained for the downstream analysis, with a median of 2242 genes and 6312 UMIs per cell (Appendix A, Appendix A). Dimensional reduction via UMAP revealed the clear segregation of cells into distinct groups (Figure 1B,C). Canonical markers and highly differentially expressed genes (DEGs) enabled us to identify six major cell types: *STAR^+^SERPINE2^+^* granulosa cells (GCs, 9614 cells, 66%) [44,45], *PTPRC^+^CD68^+^* macrophages (3601 cells, 25%), *PTPRC^+^CD1C^+^* dendritic cells (DCs, 754 cells, 5%), *PTPRC^+^CXCR2^+^* neutrophils (168 cells, 1%), *PTPRC^+^CD3D^+^CD3E^+^* T cells (260 cell, 2%) and *EPCAM^+^KRT18^+^* epithelial cells (195 cells, 1%) (Figure 1D–F).

### 3.2. GCs Constituted by Functionally Heterogeneous Subpopulations Are Involved in Different Aspects of the Ovulation Process

Granulosa cells are the primary component of the follicular microenvironment and perform various functions during the ovulation period, such as steroidogenesis, angiogenesis, oocyte meiotic resumption and so on [15,16]. There was no obvious distinction between CCs and MGCs in the GCs in the preovulatory follicles in our data (Appendix A), which was not only consistent with a previous study in rats but also the single cell results in small antral follicles in humans [46,47]. As expected, GCs in preovulatory follicles gathered in several different functional clusters. We identified nine distinct granulosa cell populations, comprising clusters G1, G2, G3, G4, G5, G6, G7, G8, and G9 (Figure 2A). To understand the functions of different GC clusters, we applied GO and KEGG analyses to identify active cellular functions and pathways in each cluster (Appendix A). 

Granulosa cells play an important role in the steroidogenesis during folliculogenesis. The major steroid hormone synthesis in GCs varies with the developmental stages of follicles. After LH/ HCG, several clusters (G1, G2 and G3) in the preovulatory follicles were all enriched with genes involved in cholesterol and steroid synthesis and metabolism, including *STAR*, *CYP11A1*, and *HSD3B2* (Figure 2B), which were important players in the P synthesis. Despite such shared characteristics, each of these clusters still exhibited unique transcriptomic patterns. G1 highly expressed several genes associated with the promotion of angiogenesis and luteotropic function, such as *ID2* and *ID4* [48]. Notably, G1 highly expressed *HSD17B1*, a classical gene involved in oestrogen metabolism (Figure 2C), indicating that G1 was not only involved in P synthesis, but also involved in E2 metabolism, promoting the conversion from E2 synthesis to P synthesis in ovaries. Compared with G2 and G3, G1 had a high level of *CYP5A* (Figure 2C). CYP5A is an alternative of NADPH-cytochrome P450 reductase to deliver the second electron to cytochrome P450 [49] and enhance the efficiency of an important non-P450 enzyme, HSD3B [50]. G3 displayed high levels of *CYP17A1*, *FDX1* and *SPARC* (Figure 2C). Among these, *FDX1* participates in the P synthesis in ovarian granulosa cells [51], and *SPARC* is involved in angiogenesis and P synthesis in the follicle-luteal transition period [52]. Therefore, these findings showed that the GCs involved in cholesterol and steroid synthesis and metabolism exhibited functional heterogeneity and can be further grouped into three distinct functional clusters.

The specific function of GCs is to closely communicate with the oocyte by providing nutritional support, transporting macromolecules throughout the folliculogenesis, and transmitting signals of LH/HCG during the ovulation. These functions are performed by intercellular connections of gap junctions between GCs and oocytes. As shown, G5 and G6 displayed high levels of *GJA1* (also knowns as *CX43*), *FN1* and *CDH2*, which are involved in cell–cell junctions (Figure 2B and Appendix A). However, the other two connexin genes, *GJC1* and *GJA5* were less expressed preovulatory GC clusters (Appendix A), which was consistent with previous studies [15,44].

More importantly, G5 highly expressed *ADAMTS1*, *ADAMTS9*, *EGR1*, *SERPINE1*, and *S100A6* (Figure 2B,D). ADAMTS1 is a secreted extensive extracellular matrix (ECM) protease and plays an important role in the ECM remodelling of the ovary tissue, a critical process for ovulation [53,54]. Granulosa cells produce versican (VCAN), which binds to the hyaluronic acid (HA)-rich matrix surrounding the cumulus–oocyte complex (COC) as well as localizes to the granulosa–thecal boundary and expanded COC matrix of preovulatory follicles [55]. Around the time of ovulation, the VCAN protein in the COC matrix is quickly cleaved by ADAMTS1 produced primarily by GCs in response to the LH surge, contributing to the tissue remodelling and follicle rupture in ovulation [13,56,57]. Here, we observed that the expression of *VCAN* fell into decline specifically in the GCs that expressed *ADAMTS1* (Appendix A), which echoed previous reports. Additionally, *ADAMTS1* is a later LH-responsive gene that can be regulated by the transcription factor, early growth response 1 (*EGR1*) [58]. *EGR1* has been reported to be induced by HCG in GCs and consequently binds to the *ADAMTS1* promotor to impact the tissue remodelling events during the periovulatory period [59]. Meanwhile, *SERPINE1*, highly expressed by G5, can inhibit the tissue-type plasminogen activator (PLAT) and, thus, reverse the suppression of cumulus expansion induced by PLAT [60]. Therefore, G5 is involved in the granulosa cellular response to LH/HCG stimulus and follicular substance remodelling (Appendix A).

G4 was characterized by the expression of *MERTK, PTGS2, PTGES, CPM* and *CNKSR3* (Figure 2B). PTGS2 (also known as COX-2) and PTGES are two rate-limiting enzymes converting the arachidonic acid into prostaglandin E2 (PGE2) [61], and are reported to drive inflammatory and apoptotic events (e.g., IL1B and CAPSs) [62] (Figure 2E) accompanying ovulation [2,63,64]. Meanwhile, the ovulation process also involves the indigestion and removal of apoptotic substrates, which requires the activation of the phagocytosis receptor MERTK [65]. The increased expression of *MERTK* in somatic GCs enables these cells to ingest the apoptotic substrates and apoptotic oocytes via an unconventional autophagy-assisted phagocytosis process [66]. Interestingly, MERTK is predicted to be regulated by PTGS2, and STAT6 signalling should be involved in this process [67,68] (Figure 2F). Therefore, G4 participates in the inflammatory responses of ovulation and the removal of apoptotic substrates, possibly by the *PTGS2/IL-1B* and *MERTK* axis. 

According to the GO analysis, G7 may participate in the positive regulation of cell migration, cell mobility, response to cytokine, focal adhesion, and chemokine activity (Appendix A). The top differentially expressed genes in G7 were *ICAM1, VCAM1, IL7R,* chemokines *CCL20*, *CXCL8*, and *CXCL1* (Figure 2B). ICAM1 and VCAM1 were noted adhesion molecules. The *CCL20* expression is regulated by the epithelial growth factor pathway, and positively correlated with the P production that dramatically increases the HCG signal [69]. The enriched expression of these genes outlines important functions of G7 in regulating cell migrations and adhesion interacting with the immune system.

In the other two GC clusters, G8 highly expressed markers associated with protein ubiquitination (such as *HERC4*, *UBR4*, *CLIP1*). G9 was characterized by the genes associated with the mitotic cell cycle and cell proliferation, such as *TOP2A*, *UBE2C*, *TTK*, *CDK1* (Figure 2B). Taken together, nine clusters of GCs in preovulatory follicles performed diverse functions to facilitate ovulation and the conversion to luteum after the LH/HCG signal. 

### 3.3. Functional Dynamics of GCs during the Luteinizing Process after HCG/GnRH-a Administration

To further understand the developmental dynamics of GC subtypes, we then performed the RNA velocity analysis. The RNA velocity analysis revealed that G1 was the initial cluster in the preovulatory GCs after the LH surge. G1 then directed to G2 and eventually to G4. We also found G5 directed to G3 and G6, which was another development branch of GCs. The latent time revealed that G5 was not the initial cluster, but it started in the middle development stage. The G3 involved in the P synthesis and the G6 involved in cell adhesion were revealed as the terminal states in preovulatory follicular GCs after the LH/HCG surge (Figure 3A,B).

LH and HCG act on preovulatory GCs to regulate the transcriptions of crucial genes necessary for ovulation and luteinization. To explore the response of some special genes to LH/HCG in humans, we explored the response time of these genes by the RNA velocity analysis. At the initial time of post-LH/HCG, the hormone synthesis in GCs transformed from the E2 synthesis to the P synthesis, which is important for luteinization. We observed that the P synthesis-related genes (such as *STAR*, *CYP11A1*, and *HSD3B2*) were upregulated earliest after LH/HCG (Figure 3C). Meanwhile, *CYP19A1*, which was involved in the E2 synthesis, occurred at the later stage but was absent at the early stage after LH/HCG, in accord with the second E2 peak in the luteal stage (Figure 3C). These changes in the steroid synthesis and metabolism were to support the luteal function, consistent with the transformation from the follicular phase to the luteal phase. The inflammation-related gene *PTGS2*, the chemokines and adhesion molecules (such as *ICAM1*, *FN1*, *CXCL8* and *CXCL1*) were activated later, whereas *CCL20* and *VCAM* were levelled up at the end of the time frame, indicating that these biological behaviours were posterior to the hormone synthesis transition (Figure 3C). Further analysis also showed that the genes involved in the regulation of oocyte meiotic maturation (such as *AREG* and *EREG*) were the latest response genes after LH/HCG (Figure 3C). These findings revealed that the transition from the E2 synthesis to the P synthesis occurs ahead of inflammatory responses and cell adhesive function, whereas the regulation of oocyte meiosis maturation is triggered at the latest (Figure 3D).

### 3.4. A Heterogeneous Macrophage Population Exists in Preovulatory Follicles with Multiple Functions

In ovaries, immune cells are present, and their quantity and functions change along the menstrual cycle. After the LH peak, immune cells infiltrate into the follicle and secrete several cytokines to assist ovulation [70]. In this study, we also observed immune cells in preovulatory follicles, including macrophages, DCs, T cells, and neutrophils (Figure 4A,B). Only two samples contained fewer NK cells; the number of NK cells in patient 1 was 18 and in patient 3 was 3. Flow cytometry confirmed the existence of CD45^+^CD68^+^ macrophages, CD45^+^CD1C^+^HLA-DR^+^ DCs, CD45^+^CD3^+^ T cells, and CD45^+^CD15^+^CD11B^+^ neutrophils in preovulatory follicles (Figure 4C). Among these, DCs and T cells expressed high levels of *RPS27*, *RPLP1*, *RPL39*, *RPL31*, *RPL23* and *RPS13* (Appendix A). The GO and KEGG pathway analyses revealed the enrichment of genes involved in ribosome, protein localization to endoplasmic reticulum, the nuclear-transcribed mRNA catabolic process and cotranslational protein targeting to the membrane (Appendix A), which are fundamental processes for immune cell activation. 

Macrophage, a dominating immune cell type in preovulatory follicles, was confirmed by immunofluorescence labelling. As shown, CD68^+^ macrophages and STAR^+^ GCs were detected in a single preovulatory follicle (Figure 5A). Macrophages exist in a wide range of organs and perform diverse functional activities to maintain tissue homeostasis [71]. We identified five clusters of macrophages (Mac1, Mac2, Mac3, Mac4, Mac5) within preovulatory follicles (Figure 5B), which are involved in a variety of functions during ovulation. 

Among these, Mac1 was characterized by the high expression of *LYVE1, RNASE1*, *DAB2, NRP1, CD163* and MRC1 (Figure 5C). *DAB2* represses the IkB kinase β-dependent (IKKβ-dependent) phosphorylation of Ser536 in the transactivation domain of *NF-kB p65* and, thus, negatively regulates the NF-kB-dependent proinflammatory gene expression [72]. *RNASE1* reduces the release of pro-inflammatory cytokines and provides multiple-organ protection in mice [73]. These findings show that Mac1 displays an anti-inflammatory and M2-like phenotype. 

Mac2 highly expressed *S100A8*, *S100A9*, *LYZ*, *CLEC4E*, and *FCN1* (Figure 5C). *S100A8* and *S100A9* mediate calcium influxes and are further involved in the state-switching of macrophages from the anti-inflammatory state (M2) to the pro-inflammatory state (M1) [74]. *FCN1* is considered a monocyte-derived pro-inflammatory macrophage marker [75], and can be secreted by macrophages [76,77]. Meanwhile, the GO analysis revealed that Mac2 was involved in various biological functions, including myeloid leukocyte activation, immune response, and exocytosis (Figure 5D). The above results indicate that this cluster is related to inflammatory responses.

Compared with other subclusters, the differential expression of genes and pathways enriched in Mac3 was down-regulated (Figure 5C,D), suggesting that this cluster may be an undifferentiated and immature population. Further studies should identify its function and differentiation direction in preovulatory follicles in IVF.

Notably, Mac4 highly expressed *CTSD*, *CTSB*, *CTSL*, *LGMN*, *APOC1*, *FABP5*, *CD9*, and *CD63* (Figure 5C). CTSD, CTSB, and CTSL belong to the cathepsin family that are proteases involved in the remodelling of ECM [78], which is essential for the follicle rupture and COC expansion. CTSD is capable of initiating a proteolytic cascade and remodelling the ECM [79]; moreover, the local accumulation of CTSB and CTSL in macrophages is associated with the degradation of tissues during inflammatory responses [80]. Furthermore, *CD9* and *CD63* are known exosome markers. *CD9* can also activate macrophages to regulate inflammatory responses [81]. Interestingly, further analysis showed CD9, CD63 and APOC1 were involved in the regulation of CTSs and MMPs [82] (Figure 5E). Therefore, Mac4, with a high secretion of exosome, plays an important role in the degradation and remodelling of the ECM (Figure 5D,E).

Differentially, the marker genes of Mac5 were associated with granulocytes and cell chemotaxis, such as *CXCL8*, *CXCL3*, *CCL3*, *CCL4*, and *CCL4L2* (Figure 5C,D). Interestingly, Mac5 also expressed genes associated with epidermal growth factors, including *AREG* and *EREG* (Figure 5C), which are important genes in the oocyte meiotic resumption after LH/HCG [7,9]. Consistent with the detected expression at the mRNA level, the results by immunofluorescence staining confirmed the expression of the EREG protein in macrophages in preovulatory follicles (Figure 5G). More importantly, CD9 and CD63 were associated with the EREG and AREG, suggesting the exosome derived by Mac4 may regulate the expression of *EREG* and *AREG* of Mac5 [83] (Figure 5F). Previous studies reported the expression of *EREG* and *AREG* in GCs and theca cells. Therefore, our study identified a cluster of macrophages that could also be a source of EREG and AREG in preovulatory follicles. Collectively, the different populations of macrophages in preovulatory follicles exhibit obvious heterogeneity in their expression profiles as well as functions. 

### 3.5. A Special G6 of GCs with High Adhesion Ability Contribute to Immune Cell Infiltration in Preovulatory Follicles

Although these six samples were all follicles with MII-stage oocytes from women who undertook IVF treatment due to male-factor infertility, the proportions of immune cells were significantly different, especially macrophages (Figure 6F). As shown, the proportions of immune cells were high in three of them (HI group) compared to the other three samples (LI group) (Figure 6A). Interestingly, there was no significant difference in the P, E2, and LH levels in these six patients right before the administration of HCG/GnRH agonist. Nevertheless, 12 h after HCG /GnRH agonist administration, the P level in the HI-group patients was significantly higher than that in the LI group, although one patient did not participate in the P level examination (Appendix A). A crucial regulator of ovulation and inflammatory responses in the ovary is P, which is induced by the LH signal, which initiates immune cell infiltration in the ovary [84]. 

The GCs in the HI group were distributed in the middle and the terminal developmental stage in the UMAP plot (Figure 6G). To further explore the potential mechanisms, we compared the DEGs of the GC populations between the two groups, which is the major cell type directly influenced by the P level. As shown, *NFκBIA* (a gene encoding NFκB inhibitor α) was lower in the GCs of the HI group than in the LI group, whereas the expression of *PTGS2* in the GCs of the HI group was higher (Figure 6B). *NFκB* is a crucial transcription factor that binds to the *PTGS2* promotor to enhance the transcriptional activity of *PTGS2* [85], which mediates inflammatory responses in ovulation and is regulated by P. Additionally, inflammatory response-related genes, such as *S100A8* and *S100A9*, were highly expressed in the HI-group GCs compared with those in the LI group (Figure 6B). In contrast, the expressions of *PTX3* and *CTSL* were significantly lower in GCs of the HI group than in the LI group (Figure 6C). PTX3 is indispensable for expanding cumulus cells to hold hyaluronan. The loss of *PTX3* prevents mouse corona radiata cells from arranging around the oocyte in preovulatory follicles, consequently influencing the expansion of COC, the remoulding of ECM, and ovulation [86]. CTSL is also important for the remodelling of ECM and ovulation [78]. Further analysis showed that the cluster G6, a population of the terminal state of GCs involved in cell adhesive functions, was predominantly located in the HI group (Figure 6D). 

Taken together, we found an obvious heterogeneity in the cell type compositions of preovulatory follicles, which was related to the different levels of macrophage adhesion and residence in follicles. Such resident macrophages in preovulatory follicles are associated with the terminal state G6 of GCs with strong adhesive ability (Figure 6D,E) as well as with enhanced inflammatory responses and poor tissue remoulding in GCs.

### 3.6. Cell Communications Contribute to the Heterogeneity in Preovulatory Follicles 

To further understand the interactions between GCs and immune cells, the cell-cell interaction analysis was performed (Figure 7A). Notably, there was a strong crosstalk between GCs and macrophages. We found that there were interactions between GCs and macrophages involved in cytokine–cytokine receptor interaction, cell adhesion molecules (CAMs), the chemokine signalling pathway and the EGFR tyrosine kinase pathway (Figure 7B). All those interactions promote macrophage chemotaxis, which then adhere and reside in the preovulatory follicle.

Additionally, the analysis of ligand–receptor interactions showed that Mac5 interacted with G2, G5 and G6 clusters of GCs via EGFR (Figure 7C). The EGFR ligand, EREG, is an EGF-like cytokine family member that plays an essential role in GCs promoting oocyte meiotic resumption. It is highly expressed in Mac5. In addition to interacting with multiple GC clusters, Mac5 also closely communicates with type 2 conventional DCs (cDC2s) via the interaction of CCL4 and its receptor CCR5, with T cells via the interaction of CCL4L2 and PGRMC2, CCL4 and CCR5, with neutrophil via the interaction of CXCL2 and CXCR2, CXCL8 and CXCR2/CXCR1, CXCL3 and CXCR2, and CCL3 and CCR1, indicating a role for macrophages in DC, T cell and neutrophil recruitment (Figure 7D).

To investigate the difference between the HI and LI groups in cellular communications, we compared the potential cell–cell interactions between these two groups. In the HI group, the *EREG* expression level in GCs was lower than that in the LI group (Figure 7E), indicating that the interactions between the Mac5 of macrophages and GC clusters via the *EREG–EGFR* axis may be stronger in the HI group. Additionally, more potential interactions between macrophage clusters and GC clusters via adhesion molecules (FN1-aVb5 complex, FN1-a5b1 complex, FN1-a4b1 complex) were observed (Figure 6E). This was consistent with the abundance of the GC cluster involved in cell adhesive functions (G6) in the HI group (Figure 6D). Therefore, the terminal state of G6 in GCs promotes the adhesion and residence of macrophages in preovulatory follicles. In turn, the infiltration and enrichment of macrophages played a critical role in the regulation of GC functions and the recruitment of DCs, T cells and neutrophils (Figure 7F).

## 4. Discussion

The microenvironment surrounding oocytes plays an important role in promoting oocyte meiotic resumption and the ovulation process. As cells from preovulatory follicles form a heterogeneous population, it has been challenging to identify the cell types and analyse their biological functions during ovulation. In this study, we presented the first single-cell investigation of the follicular microenvironment around the MII oocyte in preovulatory follicles, providing insights into the molecular mechanisms of the ovulation process. 

Several studies have utilized scRNA-seq to explore the molecular features of follicular remodelling in human ovaries [45], oocytes and granulosa cells [44,87,88]. However, these studies focused their explorations either on the transition from antral follicles to selectable follicles, or on the interactions between the oocyte and cumulus cells during folliculogenesis, leaving the preovulatory phase largely unstudied. This was mainly caused by the difficulty to obtain preovulatory follicles from human adults during the natural menstrual cycle. Here, we mapped the first single-cell transcriptomic atlas of the preovulatory follicular microenvironment around the MII-stage oocyte in IVF. We identified six major cell types in preovulatory follicles based on their unique expression markers, including a large number of immune cells. Our study not only confirmed the existence of immune cells in preovulatory follicles, but also provided a new and detailed landscape of cell types in the preovulatory follicular microenvironment, removing the barrier for further exploring the biological mechanisms underlying the ovulation process. 

As far as we know, in selectable follicles (2–5 mm in diameter), GCs can be divided into two major cell types, MGCs and CCs, with different functions [45]. Grondahl et al. [15] compared the transcriptomes of MGCs and CCs from preovulatory follicles with microarrays, which could not fully reveal the different functions within GCs. The distinctions between MGCs and CCs were not significant in preovulatory follicles, suggesting MGCs and CCs performed varieties of functions and may share some molecular functions, consistent with previous studies on ovarian follicles [46,47]. Here we identified nine different functional clusters in GCs. Three of the GC clusters were involved in steroid hormone synthesis and metabolism. G2 was probably an intermediate differentiation phenotype between G1 and G3 in terms of the expression of these key genes. It has been established that the steroid hormone synthesis in GCs transforms from the E2 synthesis to the P synthesis after the LH surge or HCG administration. Notably, our study revealed that the crucial gene involved in E2 metabolism (*HSD17B1*) was not widely enriched in all the steroid hormone synthesis and metabolism clusters, but only highly expressed in a specific cluster (G1). In addition, two other primary functions, the remodelling of the ovarian follicle ECM and the inflammatory response were identified to associate with different GC clusters. There were also GC clusters that highly expressed adhesion molecules and chemokines, which were involved in the interactions with immune cells. With the assistance of scRNA-seq, we successfully identified these functionally diverse subtypes of GCs in preovulatory follicles for the first time, suggesting that the deficiency or decrease in GC clusters may affect COC expansion, ECM remodelling, luteal function, and the inflammatory process during ovulation [56].

To gain a deep understanding of the ovulation process, it is important to recognise the cellular, hormonal and extracellular structural changes in human preovulatory follicles initiated by the LH surge or HCG administration. In this study, we found that genes with different functions of GCs play important roles at different developmental stages after LH/HCG by RNA velocity. The genes for the progesterone synthesis occur earlier than those involved in promoting oocyte meiotic resumption, inflammatory reaction and cell chemotaxis and cell adhesion. With the RNA velocity analysis, we also identified that the terminal state G6 in the development of preovulatory follicular GCs might be closer to the stage of ovulation, which was a cluster involved in cell adhesion and, thus, promoted the residence of macrophages in preovulatory follicles. 

In the GCs of preovulatory follicles, inflammatory responses, and inflammation-related gene (such as *PTGS2*, *PTGES*) expressions occur with the ovulation, right after the LH surge/HCG administration. As expected, the preovulatory follicles in our study contained an abundant population of immune cells. Our study first revealed that macrophages were the dominant immune cell type in human preovulatory follicles, which echoed a previous report in rats [84]. Interestingly, macrophages in preovulatory follicles displayed a mixture phenotype of pro-inflammation and anti-inflammation, indicating that macrophages participate in controllable inflammatory responses as well as minimal ovary tissue damage during ovulation. In particular, we identified a novel population of macrophages expressed as *EREG* and *AREG*, which were previously only known to be expressed in GCs, to assist GCs to promote oocyte maturation. Additionally, we identified a macrophage cluster that secreted protease to participate in ECM remoulding in the ovulation process and a cluster performing the recruitment of DCs, T cells and neutrophils. Therefore, macrophages probably acted as a bridge between GCs and other immune cells for promoting oocyte meiotic resumption, as well as immune responses, cell chemotaxis and ECM remoulding in ovulation (Figure 7F).

Moreover, the proportion of cell composition was different in these preovulatory follicles. Meanwhile, we found that the IVF protocols were not associated with the cell composition and the composition of GC subclusters, whereas the elevated proportion of immune cells, especially macrophages, was linked to a high cell adhesive feature of G6 in GCs, which is a terminal state in the development of preovulatory follicles. Moreover, GCs in the HI group exhibited the marked inflammatory responses and lower expression of genes necessary for the expansion of COC during the ovulation process. Combined with the RNA velocity results, the distribution of the GCs of each sample and the rapid elevation of the P level at 12 h after HCG/GnRH-a administration, the enrichment of immune cells in the HI group may occur at a relatively late stage after LH/HCG. Follicles in the HI group could be closer to a late developmental stage after LH/HCG when compared with follicles in the LI group.

As mentioned before, due to the difficulty to obtain preovulatory follicles containing M-II stage cumulus-oocyte complexes from human adults during the natural menstrual cycle in the ovary surgery, intact histological validation was not performed in our study. The variation in numbers of cells, genes and UMIs may be attributed to the different apoptosis of the GCs during the transition from the follicle phase to the luteal phase and different infiltration of immune cells [89]. Because of the small sample size and the heterogeneity of preovulatory follicles in humans, studies in larger samples and in primates could be needed in future to explore the cell type proportion and ranges of cell number in preovulatory follicular microenvironment as well as the underlying regulatory mechanisms. Our study provided a preliminary analysis of ovulatory mechanisms in preovulatory follicle and functional dynamics of GCs after LH for future clinical research. 

In conclusion, our study provides the first comprehensive single-cell transcriptomic atlas of the microenvironment in human preovulatory follicles. This work paved a path towards understanding the molecular regulation of oocyte maturation and ovulation in the preovulatory follicular microenvironment. In-vitro maturation (IVM) was a complex technique that was used for the oocyte maturation defect in IVF treatment. Macrophages are another source of EREG to promote oocyte maturation and may establish a new avenue for the development of IVM. The deficiency of genes involved in ECM remoulding and COC expansion should lead to the non-rupturing of preovulatory follicles, which will offer a new insight into the etiological diagnosis and treatment of luteinized unruptured follicle syndrome (LUFS).

## Figures and Tables

**Figure 1 biomolecules-12-00231-f001:**
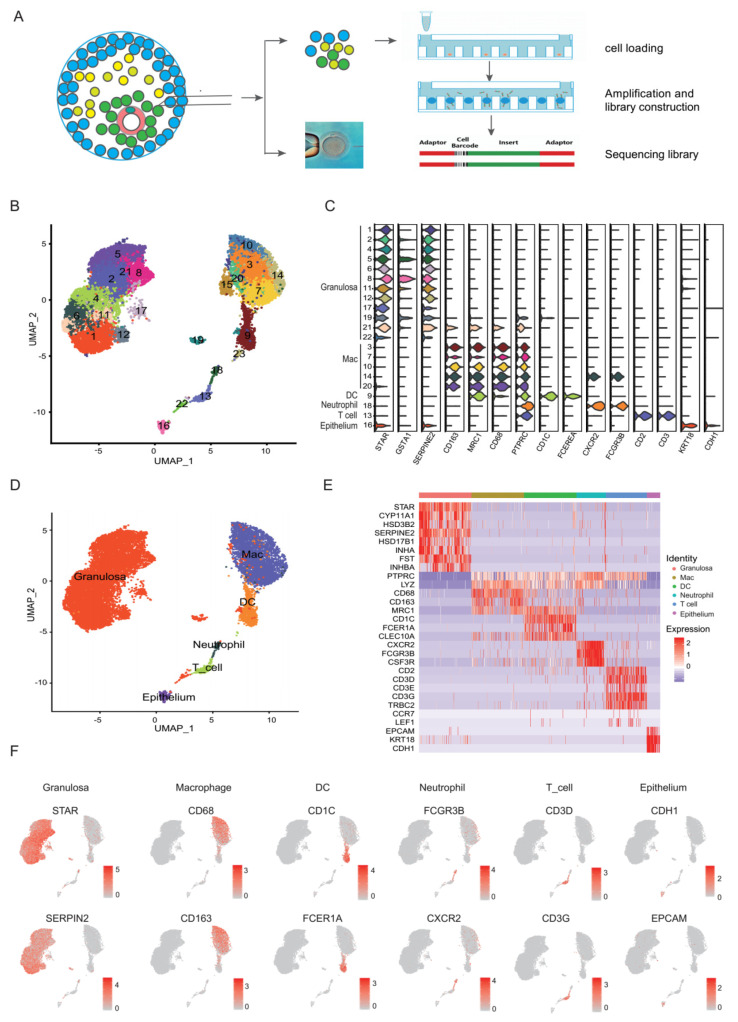
ScRNA-seq reveals multiple cell types in preovulatory follicles undergoing IVF treatment. (**A**) Schematic representation of a preovulatory follicle and the experimental procedure of the scRNA-seq study. (**B**) UMAP plot where cells that share similar transcriptome profiles are grouped by colours representing unsupervised clustering results. (**C**) Violin plots showing the expression of representative differential genes for each cluster. (**D**) UMAP plot of 14,592 high-quality cells from all samples to visualize the clustering of six cell types identified according to the expression of known marker genes, including granulosa cells (GCs), macrophages, dendritic cells (DCs), T cells, epithelial cells, and neutrophils. (**E**) Heat map showing the expression of established marker genes for each cell type in (**D**). (**F**) UMAP plots showing the expression patterns of characteristic genes in six cell types. Cells in the plots are coloured with the expression levels of the indicated genes, with a gradient from grey to red indicating the lowest to the highest gene expression level.

**Figure 2 biomolecules-12-00231-f002:**
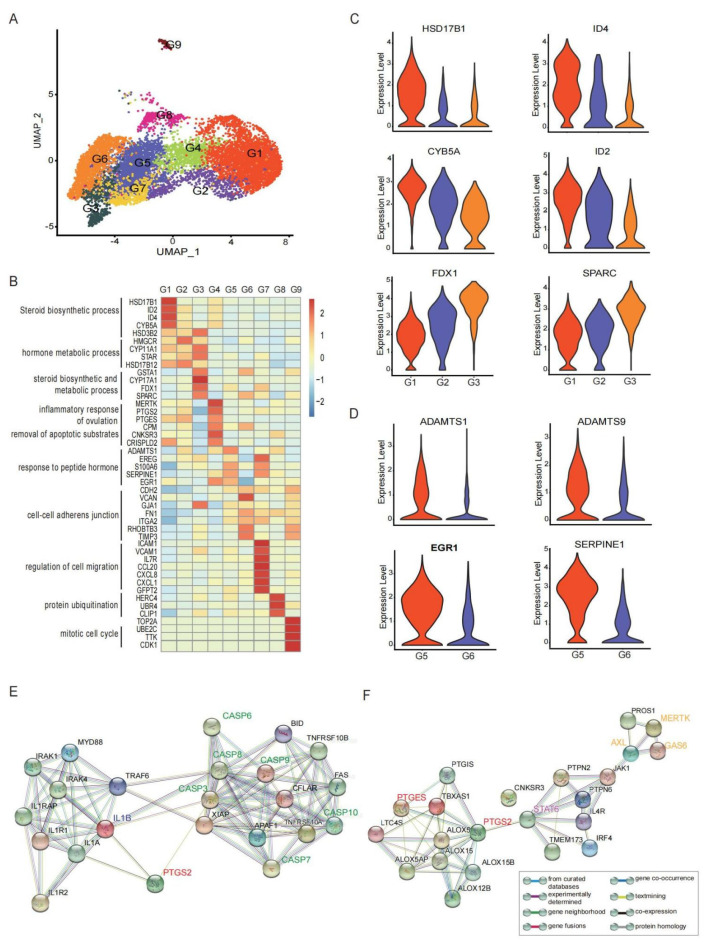
The ScRNA profile and different functional clusters of GCs in preovulatory follicles. (**A**) Nine different clusters (G1–G9) identified in the GC population, shown in a UMAP plot. (**B**) Heat map showing the expression levels of major differentially expressed genes (DEGs) in the nine GC clusters. (**C**) Violin plots showing the expression levels of major DEGs in clusters involved in the steroid synthesis and metabolism (G1–G3). (**D**) Violin plots showing the expression levels of major DEGs in GC cluster 5 and cluster 6. (**E**) Protein–protein interaction network of PTGS2, IL1B and CAPSs. (**F**) Protein–protein interaction network of PTGS2, STAT6 and MERTK.

**Figure 3 biomolecules-12-00231-f003:**
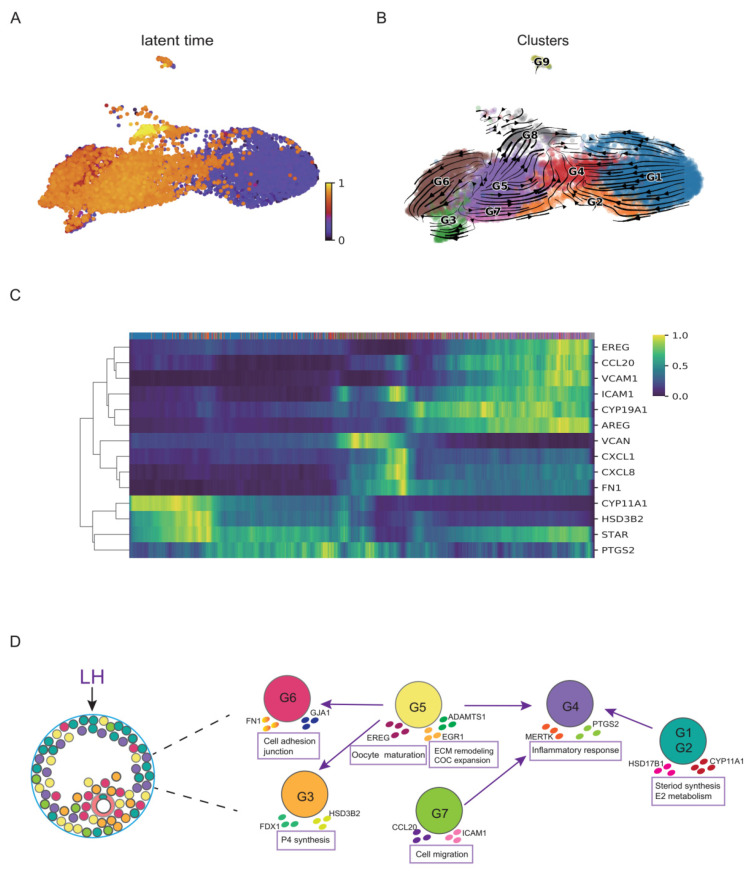
The RNA velocity and trajectory analyses of GCs. (**A**) Latent time in the RNA velocity analysis projected on the UMAP plot of the nine GC clusters. (**B**) RNA velocity analysis result of the GC clusters. Arrows show the direction of development. (**C**) The moments of spliced abundances of critical genes in GCs along the latent time, with a gradient from blue to yellow indicating the lowest to the highest normalized spliced abundance. (**D**) Diagram of the developmental dynamics of the major clusters of GCs.

**Figure 4 biomolecules-12-00231-f004:**
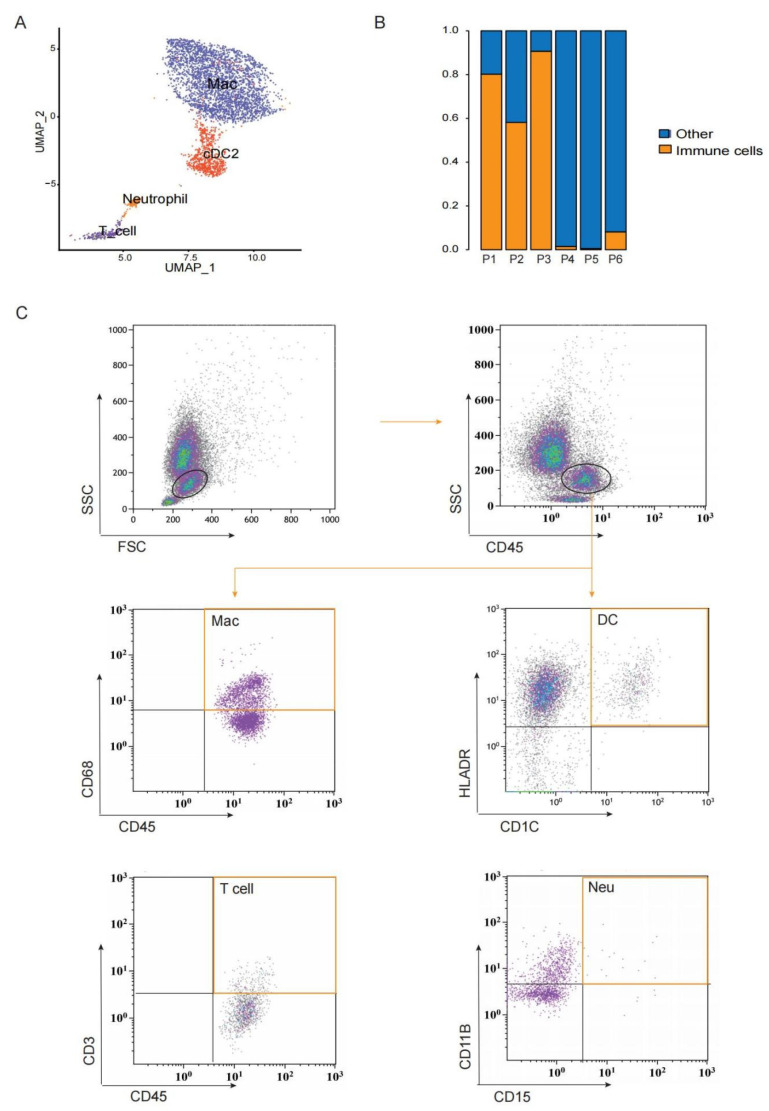
Immune cells exist in preovulatory follicles. (**A**) UMAP showing immune cell types in preovulatory follicles. (**B**) The proportion of immune cells in each preovulatory follicle sample. (**C**) Flow cytometry analysis identifying CD45^+^ immune cells, macrophages (sorted with CD45 and CD68), DCs (sorted with CD1C and HLA-DR), T cells (sorted with CD45 and CD3) and neutrophils (sorted with CD11B and CD15) in a single preovulatory follicle.

**Figure 5 biomolecules-12-00231-f005:**
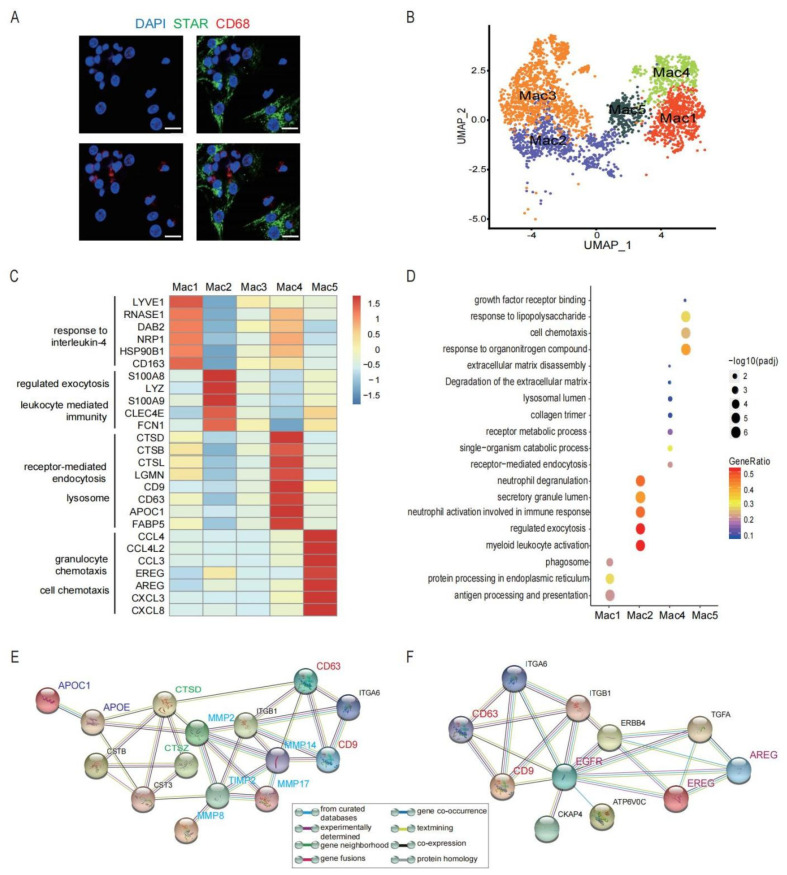
Five functionally different clusters identified in macrophages in preovulatory follicles. (**A**) Immunofluorescence showing the existence of GCs (STAR, green) and macrophages (CD68, red) in preovulatory follicles. DAPI (blue)-labelled cell nucleus. Scale bar, 20 um. (**B**) Five different clusters (M1–M5) identified in the macrophage population, shown in a UMAP. (**C**) Heat map showing the expression levels of major DEGs in the five macrophage clusters. (**D**) Bubble plot showing the representative GO terms enriched in the five macrophage clusters. (**E**) Protein–protein interaction network of CD9, CD63, CTSs and MMPs. (**F**) Protein–protein interaction network of EREG, AREG, CD9 and CD63. (**G**) Immunofluorescence showing EREG protein expression (red) in macrophages in preovulatory follicles. CD68 fluorescence (green)-labelled macrophages, and DAPI (blue)-labelled cell nuclei. Scale bar, 25 um. Immunofluorescence showing DAB2, S100A8, and CTSD protein expression (green) in macrophages in preovulatory follicles. CD68 fluorescence (red)-labelled macrophages, and DAPI (blue)-labelled cell nuclei. Scale bar, 20 um.

**Figure 6 biomolecules-12-00231-f006:**
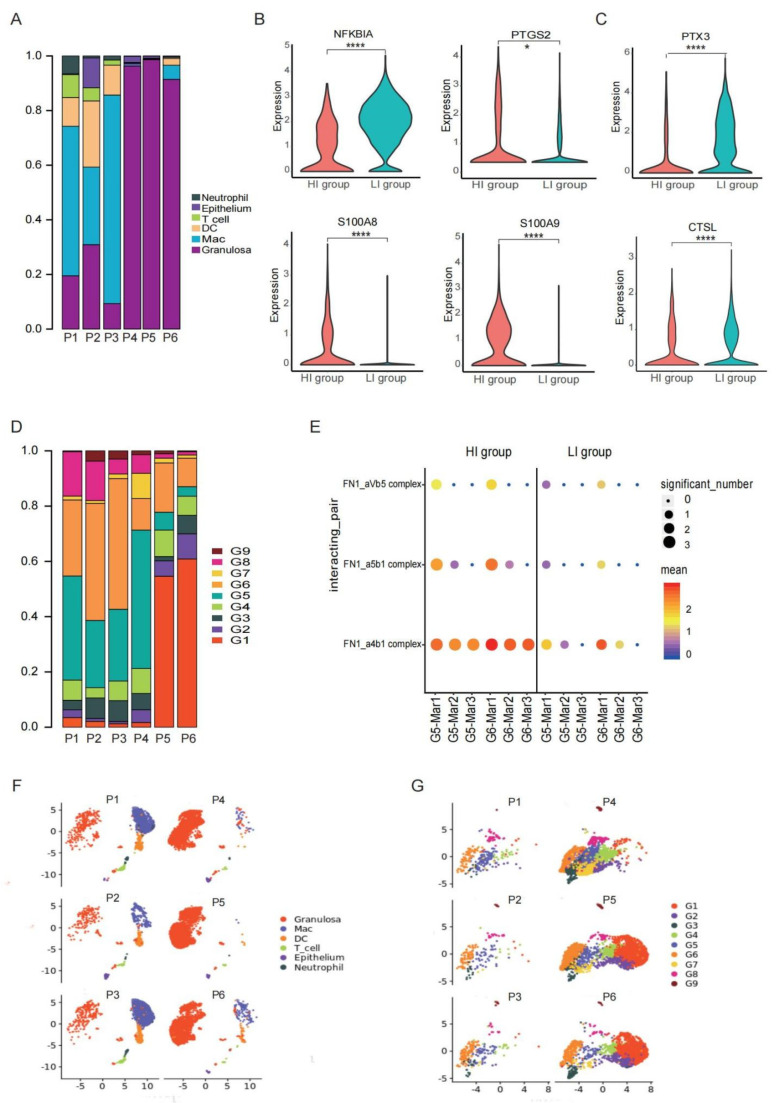
Intrinsic heterogeneity in preovulatory follicles. (**A**) Cell type compositions in each of the six preovulatory follicle samples. (**B**) Violin plots showing the expression levels of NFkBIA, PTGS2, S100A8 and S100A9 in GCs of the HI and LI groups. **** *p* < 0.0001 and * *p* < 0.05 (Wilcox test). (**C**) Violin plots showing the expression levels of PTX3 and CTSL in GCs of the HI and LI groups. **** *p* < 0.0001 (Wilcox test). (**D**) Stacked bar graph showing the composition of GCs in each sample. (**E**) Bubble plot showing the ligand-receptor interactions mediated by adhesive molecules between GCs and macrophages in the HI and LI groups. (**F**) UMAP plot showing the individual distribution of all cells. (**G**) UMAP plot showing the individual distribution of all GCs.

**Figure 7 biomolecules-12-00231-f007:**
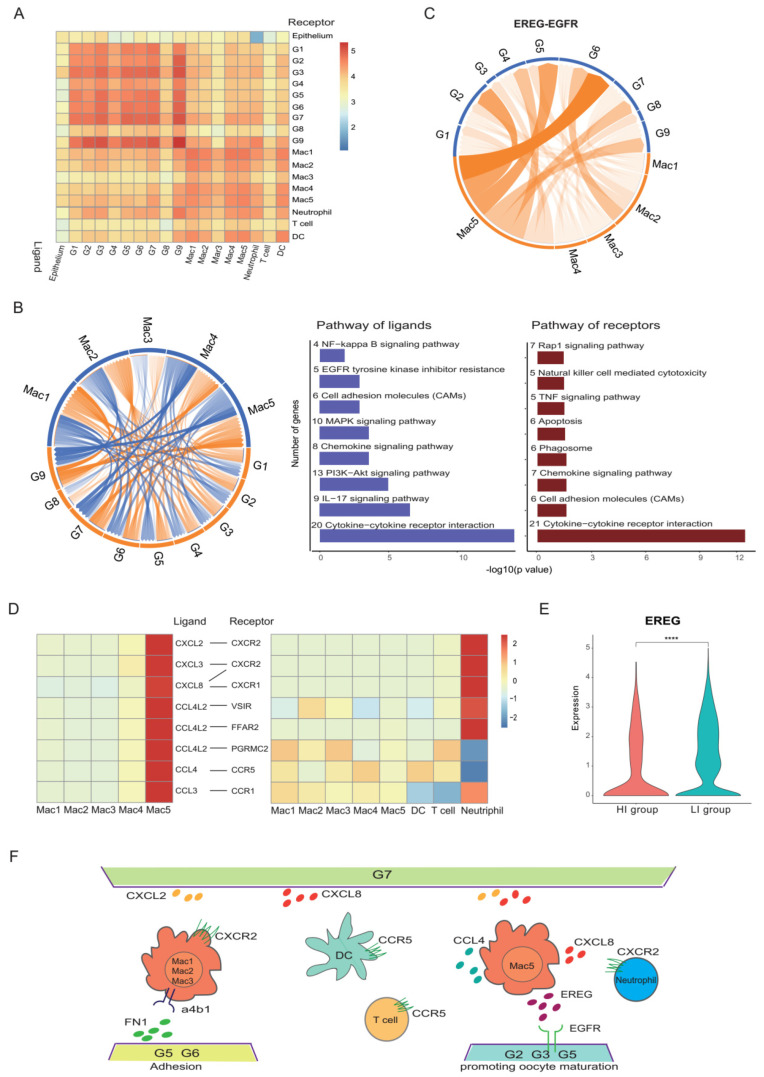
Cell-cell interactions in preovulatory follicles. (**A**) Heat map showing the interactions between different cell types. (**B**) Circos plot showing the interactions between GCs and macrophages (left). The ligand–receptor interaction pathway of GCs and macrophages (right). (**C**) Circos plot showing the interactions between EREG on macrophage clusters (orange) and EGFR on GC clusters (blue). The thickness and darkness of the connecting lines in the middle of the circle indicate the prevalence of the interaction in the six samples. (**D**) Heat map showing the chemokine interactions between macrophages, DCs, T cells and neutrophils. (**E**) Violin plots showing the expression levels of EREG in the GCs of the HI and LI groups. **** *p* < 0.0001 (Wilcox test). (**F**) Diagram of the main ligand and receptors’ interaction of GCs and immune cells.

## Data Availability

The data presented in this study have been uploaded on the GEO database. The GEO accession number is GSE189960.

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
