# Peer review of "Single-Cell Sequencing Reveals an Intrinsic Heterogeneity of the Preovulatory Follicular Microenvironment"

_biomolecules, 2022, doi:10.3390/biom12020231_

Round 1
Reviewer 1 Report
The authors describe generally well their work in single cell sequencing and analysis of data on preovulatory follicular microenvironment and the work is a good initial step in describing the composition and roles of different cell populations in this environment. It is also excellent example that the data has been submitted to a public international repository. However, I feel that there are several points that need to be improved in the manuscript.
- The introduction contains a lot of information that may be on the borderline general knowledge in the reproduction field but in a general journal more references for this information should be provided, maybe a figure could be included to describe the follicular microenvironment and some acronyms should be opened (most already are but not all).
- The primary analysis of data requires more detailed description. How were the UMIs counted for example?
- The quality control of the data is a bit confusing? Why did the authors allow such a high threshold for mitochondrial gene expression? Experimental comparison is not enough to justify such a choice. The authors should also comment why top 20 PCs were used and why the resolution parameter was chosen to be 1.2.
- In the results, the authors should report the sequencing results in detail. No only the results after QC are reported.
- The results from single follicles should be presented and discussed instead of only combining the samples. We see that there is for example a significant difference in the amount of immune cells between the samples.
- The results section contains a lot of speculative or suggestive discussion ("these findings suggest...", "are probably...") which is not supported by any references to litterature. I believe that all these parts should be moved to discussion and supported by some related previous work. Currently, the are mixed with actual results and purely authors speculation.
- The authors discuss the current view on composition of GCs but do not really reflect their results on this. How do the results relate to CCs and MGCs? What are the cell types that the authors have identified?
- Even though the language of the manuscript is generally very good, some grammatical errors that deal with sentence structure and make it harder to read in places should be corrected.
Author Response
Reviewer(s)' Comments to Author:
Review 1:
The authors describe generally well their work in single cell sequencing and analysis of data on preovulatory follicular microenvironment and the work is a good initial step in describing the composition and roles of different cell populations in this environment. It is also excellent example that the data has been submitted to a public international repository. However, I feel that there are several points that need to be improved in the manuscript.
- The introduction contains a lot of information that may be on the borderline general knowledge in the reproduction field but in a general journal more references for this information should be provided, maybe a figure could be included to describe the follicular microenvironment and some acronyms should be opened (most already are but not all).
Responses: Thank you for the suggestion. We have supplemented the description of follicular microenvironment in the section of Introduction. We also added the references for the information in the reproduction field and the follicular microenvironment (References number 2-6, 13-14) in the section of Introduction. The leftmost figure in Figure 1A described the preovulatory follicular microenvironment containing cumulus cells and the cells in the follicular fluid.
- The primary analysis of data requires more detailed description. How were the UMIs counted for example?
Responses: Thank you. We have supplemented the description of the primary analysis of data (Line 128-139).
- The quality control of the data is a bit confusing? Why did the authors allow such a high threshold for mitochondrial gene expression? Experimental comparison is not enough to justify such a choice. The authors should also comment why top 20 PCs were used and why the resolution parameter was chosen to be 1.2.
Responses: Thank you. We compared different thresholds for mitochondrial gene expression (50% vs 20%) and the result showed there are very minimal changes in the cells removed, which also demonstrated good data quality of this dataset. The detailed information of each sample was supplemented in Table S2.
Based on the literature and experience, top 20 PCs are usually enough to capture the variances between major cell types in the high dimensional space. We also used the elbow plot and confirmed that at the 20th PC, the standard deviation curve already flattens out.
We chose the resolution to be high enough (1.2) to ensure major cell types well separated. In the following annotation step, clusters annotated as the same cell type were combined into annotated cell types and processed into subtype clustering.
- In the results, the authors should report the sequencing results in detail. No only the results after QC are reported.
Responses: Thank you for the suggestion. The detailed sequencing results were added to the Results section (Line 229-231) and the detailed information of each sample before and after quality control were supplemented in Table S2. We modified the results after QC from the average (mean) genes and UMIs to median genes and UMIs (Line 232).
- The results from single follicles should be presented and discussed instead of only combining the samples. We see that there is for example a significant difference in the amount of immune cells between the samples.
Responses: Thank you. The UMAP plot distribution of individual samples in all cells and GCs were added in Figure 6E and 6F. We supplemented the discussion of these differences in the discussion section (Line 626-630).
- The results section contains a lot of speculative or suggestive discussion ("these findings suggest...", "are probably...") which is not supported by any references to litterature. I believe that all these parts should be moved to discussion and supported by some related previous work. Currently, the are mixed with actual results and purely authors speculation.
Responses: Thank you for the suggestion. We have moved these parts of results to the discussion section.
- The authors discuss the current view on composition of GCs but do not really reflect their results on this. How do the results relate to CCs and MGCs? What are the cell types that the authors have identified?
Responses: Thank you for the suggestion. MGCs and CCs are distinguished by their location in the follicle. The GCs around the oocyte in the cumulus-oocyte complexes are CCs while the others are MGCs. We found no obviously distinction between CCs and MGCs in the GCs in the preovulatory follicles, which was consistent with previous study in rat and the single cell results in human small antral follicles. (1) Szoltys M, Slomczynska M, Knapczyk-Stwora K, et al. Immunolocalization of androgen receptor and steroidogenic enzymes in cumuli oophori of pre- and post-ovulatory rats[J]. Acta Histochem. 2010, 112(6): 576-582. (2) Fan X, Moustakas I, Bialecka M, et al .Single-Cell Transcriptomics Analysis of Human Small Antral Follicles. Int J Mol Sci. 2021, 22(21). Our study suggested that MGCs and CCs performed varieties of functions and may share some molecular functions. We have supplemented the results about MGCs and CCs in the results section (Line 255-258) and the discussion section (Line 575-578).
- Even though the language of the manuscript is generally very good, some grammatical errors that deal with sentence structure and make it harder to read in places should be corrected.
Responses: Thank you. We have checked the manuscript and revised grammatical errors, and it was edited by a professional editing service (https://mdpi.cn/about/english-editing).

Reviewer 2 Report
Results in the study performed by Wu et al. are interesting and adding information to preovulatory follicle cell niche. However, additional confirmatory experiments are required to support the conclusion.
Do authors are able to present further developmental competence of MII oocytes isolated from analysed follicles (Fig. 1A)?
Presented data in the Fig. 2EF & 5G are not explanatory. Adaptation of data presentation from e.g. PMC4123855 would be beneficial here.
Please provide information about viability analysis of the cells prior to loading to microfluidic devices.
Fig. 4B shows significant inconsistency between the samples. One can speculate that observed diverse cell types might be simply result of contamination during oocyte retrieval. Authors show expression of various markers in the cell retrieved from follicle, however the data should be validated on the histological samples to maintain follicle integrity.
Fig. 4B Color-coding appears different between P1 and rest of patients.
Fig. 5G Scale bars are invisible.
Author Response
Review 2:
Results in the study performed by Wu et al. are interesting and adding information to preovulatory follicle cell niche. However, additional confirmatory experiments are required to support the conclusion.
Do authors are able to present further developmental competence of MII oocytes isolated from analysed follicles (Fig. 1A)?
Responses: Thank you. The developmental competence of MII oocytes were in the embryo score in the last row of Table 1.
Presented data in the Fig. 2EF & 5G are not explanatory. Adaptation of data presentation from e.g. PMC4123855 would be beneficial here.
Responses: Thank you for the suggestion. We have supplemented related studies to support the information (References number 50, 55-56, 70-71).
Please provide information about viability analysis of the cells prior to loading to microfluidic devices.
Responses: Thank you. We have supplemented the information about viability analysis of the cells in Table S2.
Fig. 4B shows significant inconsistency between the samples. One can speculate that observed diverse cell types might be simply result of contamination during oocyte retrieval. Authors show expression of various markers in the cell retrieved from follicle, however the data should be validated on the histological samples to maintain follicle integrity.
Responses: Thank you for the suggestion. Due to the ethical reasons, intact human follicle tissue cannot be obtained in in vitro fertilization. We can hardly accurately acquire the intact human follicle containing MⅡ-stage cumulus-oocyte complexes in the ovarian surgery because of we cannot control the ovulation in natural menstrual cycle. In order to reduce the impact of contamination, we choose the first punctured follicle and ensured that the follicular fluid was clear and free of visible blood contamination. We supplemented these in the methods section (Line 106-107).
Fig. 4B Color-coding appears different between P1 and rest of patients.
Responses: Thank you. We have revised the color of P1 in Fig. 4B.
Fig. 5G Scale bars are invisible.
Responses: Thank you. We have adjusted the scale bar in Fig. 5G.
Round 2
Reviewer 1 Report
The authors have properly addressed most of the points raised in the first review, although it has been a bit difficult to follow this as the line numbers in the provided manuscript and response letter are not consistent. However, there still remain some issues that need clarification and additional data added as a result of authors recent additions raises new questions:
- All the used software tools and computational methods should be accompanied by an appropriate citation.
- The authors should also describe the sequencing depth as this is relevant for assessing the data quality
- Table S2 shows the metrics for different samples. There are large differences in all the shown metrics, even after quality control. The authors should discuss the effect of viability and especially the large differences in cell numbers, detected UMI and gene counts and percent of mitochondrial genes.
- In the light of Table S2 the quality control performed by the authors seems inadequate. The authors should justify why they accept so high percent of mitochondrial genes instead of e.g. 5%. The same goes with the number of genes and UMIs. It is possible to present sample-wise violin plots for these features for example to justify the use of these limits but all three filtering parameters seem to be very high in their upper limits. This naturally leads to retaining more cells but may allow erroneous cells to be included in the analysis. With these parameters, there are very few cells filtered out. Is there a specific reason to believe that the used platform does not contain duplicate cells or other erroneous "cells" which is usually the case with other platforms such as 10x? It would be good to see the effect of more usual, stricter quality filtering limits to the number of cells and subsequent results. It is also not very productive to report average numbers over all the cells as we can see from the Table S2 that there are big differences in the cell numbers and gene/UMI numbers which causes the samples with more cells and larger numbers to dominate the averages.
- The authors have provided per-sample version for their UMAP-plots. However, partly due to the colors the authors have chosen, it is very hard to distinguish some of the samples and their contribution from the plots. In the light of the numbers presented in Table S2, I would suggest the authors show each sample separately in a similar fashion to marker gene expression which allows the assessment of each sample separately. With the big differences in the cell numbers and proportions in GC and immune cell populations between the samples, there is a concern that for example in P3 there are very few GC cells that are further separated to different clusters. It would be good to report the number of different cell types for each sample to allow transparent analysis of the results and to assess the support for conclusions.
- There are now two E subfigures in Figure 6
Author Response
The authors have properly addressed most of the points raised in the first review, although it has been a bit difficult to follow this as the line numbers in the provided manuscript and response letter are not consistent. However, there still remain some issues that need clarification and additional data added as a result of authors recent additions raises new questions:
1. All the used software tools and computational methods should be accompanied by an appropriate citation.
Responses: Thank you. We have added the references for the software tools and computational methods in the Material and methods section (Reference number 24-27, 31-33, 35, 38-39).
2. The authors should also describe the sequencing depth as this is relevant for assessing the data quality
Responses: Thank you. We have supplemented the sequencing depth (mean reads per cell) in the Table S2.
3. Table S2 shows the metrics for different samples. There are large differences in all the shown metrics, even after quality control. The authors should discuss the effect of viability and especially the large differences in cell numbers, detected UMI and gene counts and percent of mitochondrial genes.
Responses: Thank you. Because of our samples were from a single follicle containing preovulatory follicular fluid and the cumulus cells, the cell numbers were quite different in individuals. This may mainly be caused by the transition from follicle phase to luteal phase after LH/HCG and this transition is accompanied by GCs apoptosis (Sugino N, et al. Nitric oxide concentrations in the follicular fluid and apoptosis of granulosa cells in human follicles. Human Reproduction 1996. 11:2484-2487). Even though we acquired the samples at the same time after LH/HCG, the different lutenized stage of GCs was found in our study and then the apoptosis of GCs were different in individuals. As for immune cells, owing to the different stage before ovulation, the infiltration of immune cells were quite different, which was consistent with previous study by Flow Cytometry ( Smith M P, et al. Leukocyte origin and profile in follicular aspirates at oocyte retrieval. Hum Reprod. 2005, 20(12): 3526-3531). We have supplemented the explanation and limitation in the discussion section (Line 643-651).
4. In the light of Table S2 the quality control performed by the authors seems inadequate. The authors should justify why they accept so high percent of mitochondrial genes instead of e.g. 5%. The same goes with the number of genes and UMIs. It is possible to present sample-wise violin plots for these features for example to justify the use of these limits but all three filtering parameters seem to be very high in their upper limits. This naturally leads to retaining more cells but may allow erroneous cells to be included in the analysis. With these parameters, there are very few cells filtered out. Is there a specific reason to believe that the used platform does not contain duplicate cells or other erroneous "cells" which is usually the case with other platforms such as 10x? It would be good to see the effect of more usual, stricter quality filtering limits to the number of cells and subsequent results. It is also not very productive to report average numbers over all the cells as we can see from the Table S2 that there are big differences in the cell numbers and gene/UMI numbers which causes the samples with more cells and larger numbers to dominate the averages.
Response: Thank you for your suggestion. In the follicular microenvironment, glucose is metabolized in GCs and almost none in the oocyte. Glucose metabolism is essential for meiotic maturation of oocytes as it is used to generate ATP, purines and nucleic acids for DNA synthesis, NADPH for redox homeostasis and hyaluronic acid for extracellular matrix formation during cumulus expansion. As mentioned in previous studies, glucose consumption in GCs progressively increases throughout oocyte maturation in order to meet the increasing demand for energy and metabolites that support COC maturation. ((1) Steeves TE, et al. Metabolism of glucose, pyruvate, and glutamine during the maturation of oocytes derived from pre-pubertal and adult cows. Mol Reprod Dev 1999;54:92–101. (2) Sutton ML, et al. Effects of in-vivo and in-vitro environments on the metabolism of the cumulus-oocyte complex and its influence on oocyte developmental capacity. Hum Reprod Update 2003;9:35–48). Moreover, the production of enzymes involved in steriod synthesis, such as STAR, CYP11A1, increases in GCs in order to support luteal function after LH/HCG. These enzymes are involved in cholesterol conversion in mitochondria. Therefore, we chose a relatively high mitochondria gene percentage to retain the GCs with high metabolic activity in preovulatory stage, which was also a major cell type in preovulatroy follicle.
As for detected genes and UMIs, the upper limits are suggested to remove potential doublets. There is scarcely any scRNA-seq research on the preovulatory follicles in human, so we don’t want to remove some real existed cells with a stricter filtering limit. The technology itself has a very wide dynamic range of detected genes and UMIs. In order to ensure the quality of all retained and analyzed cells, we later removed annotated doublets and low-quality cells at the annotation step. We have supplemented the sample-wise violin plots of these features in Supplementary figure 1 (Figure S1) . We modified the description from average number to median for genes and UMI in the Material and methods section 2.5 (Line 147-148) and explained the filter criteria (Line 143-147, 165-167).
5. The authors have provided per-sample version for their UMAP-plots. However, partly due to the colors the authors have chosen, it is very hard to distinguish some of the samples and their contribution from the plots. In the light of the numbers presented in Table S2, I would suggest the authors show each sample separately in a similar fashion to marker gene expression which allows the assessment of each sample separately. With the big differences in the cell numbers and proportions in GC and immune cell populations between the samples, there is a concern that for example in P3 there are very few GC cells that are further separated to different clusters. It would be good to report the number of different cell types for each sample to allow transparent analysis of the results and to assess the support for conclusions.
Response: Thank you for the suggestion. We have revised per-sample version for the UMAP plot. The UMAP plot distribution in all cells and GCs were presented individually with each sample in Figure 5F and Figure 5G. We have added the numbers of different cell type for each sample in Table S3.
6. There are now two E subfigures in Figure 6
Response: Thank you. We We apologized for the carelessness and have revised the subfigures to F and G in Figure 6.
Reviewer 2 Report
Table S1 need expand list of abbreviation used. Why patient was changed to P and P is abbreviated as progesterone? How embryo outcome correlates with results?
Table S2 need comments in the M&M and text. How viability was tested?
The text e.g. 2.1 or 3.1 doesn’t make sense, IVF or ICSI?
Authors show expression of various markers in the cell retrieved from follicle, however the data should be validated on the histological samples to maintain follicle integrity. Based on your response discuss obtained data and validation.
Author Response
Table S1 need expand list of abbreviation used. Why patient was changed to P and P is abbreviated as progesterone? How embryo outcome correlates with results?
Response: Thank you. We apologized for the carelessness of the change and we have restored the name of patient in the Table S1. We have supplemented the abbreviation in Table S1. The oocyte was fertilized by ICSI and developed into embryo. The embryo score indicated the quality of the embryo and high quality embryo was defined as embryos with 7-9 cells and <30% fragmentation on day 3 or blastocysts meet or exceed 3BB by Gardner score. Therefore, the embryo score of 3BB and 4BB in the Table S1 were high quality embryos. Conversely, the embryo score of 1 cell and 332 in Table S1 were poor quality embryos and could not reach the blastocyst stage. We have supplemented the scoring criteria in the annotation.
Table S2 need comments in the M&M and text. How viability was tested?
Response: Thank you. Trypan blue exclusion test was used to evaluate the cell viability. The percentage of cells that have clear cytoplasm (viable cells) was viable cell percentage. We have supplemented the content in the section of Material and methods and the text (Line 117-118).
The text e.g. 2.1 or 3.1 doesn’t make sense, IVF or ICSI?
Response: Thank you. All the oocytes from the six patients were fertilized by ICSI because of the male factor. We have revised in the section of Material and methods and Results.
Authors show expression of various markers in the cell retrieved from follicle, however the data should be validated on the histological samples to maintain follicle integrity. Based on your response discuss obtained data and validation.
Response: Thank you for the suggestion. We have supplemented the explanation in Discussion section (Line 640-643).